# Coordinated hippocampal-entorhinal replay as structural inference

**Talfan Evans**
Institute of Cognitive Neuroscience
University College London
talfan.evans.13@ucl.ac.uk

**Neil Burgess**
Institute of Cognitive Neuroscience
University College London
n.burgess@ucl.ac.uk

## Abstract

Constructing and maintaining useful representations of sensory experience is essential for reasoning about ones environment. High-level *associative* (topological) maps can be useful for efficient planning and are easily constructed from experience. Conversely, embedding new experiences within a *metric* structure allows them to be integrated with existing ones and novel associations to be implicitly inferred. Neurobiologically, the synaptic associations between hippocampal place cells and entorhinal grid cells are thought to represent *associative* and *metric* structures, respectively. Learning the place-grid cell associations can therefore be interpreted as learning a mapping between these two spaces. Here, we show how this map could be constructed by probabilistic message-passing through the hippocampal-entorhinal system, where messages are scheduled to reduce the propagation of redundant information. We propose that this *offline* inference corresponds to co-ordinated hippocampal-entorhinal replay during sharp wave ripples. Our results also suggest that the *metric* map will contain local distortions that reflect the inferred structure of the environment according to *associative* experience, explaining observed grid deformations.

## 1 Introduction

Localizing in an environment relies on two sources of information. Firstly, unique sensory inputs may indicate absolute location in space. Secondly, path integration (PI) can update previous location on a *metric* map by integrating self-motion. Sensory inputs are required to correct the accumulation of error by PI, but problems arise when their role in localization occurs simultaneously with learning of their correspondence to locations on the *metric* map (SLAM) [10]. In general, computing the joint *map-location* distribution requires probabilistic inference over previous sensory observations and movements given their respective uncertainties. *Associative* representations can be computationally cheaper when used to perform high-level planning [56]. However, organizing *associative* structure in a *metric* space allows for efficient integration of new experience and the inference of *metric* relationships between sensory states in the absence of physical experience. This 'short-cutting' ability is crucial for efficient exploration and navigation [58; 55].

### 1.1 Place and grid cells

Neurobiologically, grid cells (GC) in the medial entorhinal cortex (mEC), whose firing fields are arranged on a periodic hexagonal lattice in space, are thought to play a role in PI [17] and constitute a *metric* map of space [22]. Their firing patterns are stable over time suggesting stabilization by environmental cues [24; 12]. Conversely, place cells (PC) in the hippocampus (HPC) fire at distinct locations [43] and are thought to respond to specific sensory stimuli such as environmental geometry [44; 30]. PCs represent states in sensory space such that their activity most often reflects the animal's

current location, their synaptic associations constitute an *associative* map of an environment [40] and their connections to GCs stabilize the GC *metric* map. Although PC and GC activity most often represents the current location, coordinated sequential 'replay' of remote cells (i.e. whose firing fields are non-local) also occurs [15; 45].

## 1.2 Summary of contributions

This work proposes a novel dual-systems account of probabilistic localization and learning in the HPC-mEC system, on both an algorithmic and implementation level [33]. Predictions from our hypothesis are evaluated by comparison to existing experimental data by both numerical simulations and theoretical analyses.

We propose that the HPC-mEC system operates in two distinct regimes. When navigating using a known *map* (i.e. locations of sensory states in *metric* space), an *online* system probabilistically integrates PI and sensory information for localization (Fig. 4A). A simple learning mechanism allows the *online* system to learn initial priors over the *map* structure (Fig. 4B). However, conflicts in the PI and sensory estimates of location necessitate more complex *offline* inference to correct the *map*, which requires inference over previous sensory observations and movements (Fig. 2) [48].

We show how this *offline* system can use the *associative* structure stored in the recurrent CA3 synapses between PCs to construct and correct a *metric* map stored in the synaptic associations between GCs and PCs, corresponding to one-shot learning. The distribution over landmark locations is computed via message passing [47] between PCs. Scheduling messages to minimize the propagation of redundant information not only improves performance, but also produces structured reactivations of PCs resembling those observed during hippocampal 'replay' [15; 9; 50]. Our model provides both a functional and mechanistic interpretation for observations of coordinated HPC-mEC replay [45; 64] and makes novel experimental predictions. In contrast to reward-based interpretations [36], our model poses replay as structured coordinated information transfer though complementary *metric* and *associative* representations of the world. Sharp-wave ripples [7; 42] which coincide with replay events may correspond to structural prediction errors.

Lastly, when the learned *associative* structure is non-Euclidean, organization within a *metric* space predicts recently observed local distortions in GC firing patterns, such that the underlying structure represented by GCs reflects the informational 'similarity' of distinct locations in stimulus space.

## 2    Related work

Previous theoretical models have proposed how current location might be represented in the population firing rates of grid cells [35], updated by PI information and corrected by sensory inputs [24; 17; 13]. However, these mechanisms do not describe how uncertainty in either the PI or sensory inputs could be integrated probabilistically, instead assuming that input from familiar landmarks 'reset' the current distribution.

Probabilistic localization assumes an existing mapping from sensory stimuli to location in *metric* space. Although several studies have demonstrated how this mapping might be learned [39; 38; 41], these models do not link experimentally observed local distortions [23; 54] in the firing patterns of grid cells to non-uniformities in their underlying stimulus input, or to non-uniform behavioural sampling.

Neither do these models link to the phenomenon of coordinated HPC/mEC replay. During replay, place and grid cells with overlapping spatial firing fields are observed to reactivate during *offline* states (such as sleep, grooming or pausing at choice points), in spatial sequences that recapitulate behavioural trajectories experienced by the animal during *online* behaviour [16; 9]. Forward [9] and reverse [16] is thought to be associated with planning and consolidation during reward-based learning, respectively. Of existing models, only one provides normative insights [37] and none account for coordinated mEC-HPC replay [29; 46; 64]. Ours is the first model to implicate replay in probabilistic learning of spatial structure, providing an alternative view to reward-based accounts.

Our model does not tackle the problem of learning the form of the metric mEC space into which the environmental structure is embedded, although it is possible that these representations can emerge

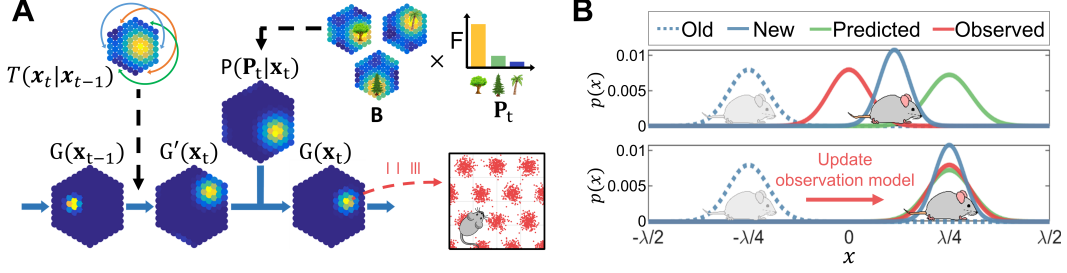

Figure 1: **A** The *online* model. The GC firing rates at time $t$ are updated by PI before correction by weighted input from PC firing. Each hexagon defines a single grid module with $N_G$ GCs. Plotting spikes from a single GC against the position of the animal generates the GC firing pattern. **B** Learning corrects the *observation model* towards the *predicted estimate*.

from unsupervised learning in navigational tasks [2; 62] or as the eigendecomposition of the transition matrix between states in an environment [52] or as predictive functions of sensory inputs [61].

# 3 Model

## 3.1 Grid and place cells

A single GC will fire periodically at the vertices of a triangular lattice in 2D space (Fig. 1A). GCs exist in anatomical 'modules', groups of GCs whose firing patterns share the same spatial scale (distance between vertices) and orientation relative to the environment, but differ in their spatial offsets [22]. Moreover, the spatial scale increases in discrete 'jumps' along one anatomical axis of the mEC, suggesting that these modules encode a hierarchical representation of space [3; 14; 34]. $G(\mathbf{x})$ describes the probability distribution over current location within a periodic, discretized region of state space $\mathbf{x}$. Biophysically, this would be represented by the firing rates of $N_G$ GCs $\mathbf{G}$ (i.e. a discretization over the support of $G(\mathbf{x})$). Although we only consider a single grid scale, our results naturally extend to multi-scale architectures, theoretically allowing encoding of ranges up to [21; 34; 60] or beyond [14] the largest grid scale.

PC firing $\mathbf{P}$ represents the probability of the presence of specific sensory stimuli, which in a spatial context can be considered as 'landmarks' whose location in physical space is denoted by $\boldsymbol{\mu}_p$. In our simulations, the firing of each of the $N_P$ PCs is described by a Gaussian receptive field $p_t^p \sim f(\hat{\mathbf{x}}_t'|\boldsymbol{\mu}_p, \sigma_{\text{PC}}^2\mathbf{I})$, where $\hat{\mathbf{x}}_t' \sim \mathcal{N}(\mathbf{x}_t', \sigma_{\text{PC}}^2\mathbf{I})$ is a noisy estimate of the current position in physical space $\mathbf{x}'$. We will use the notation $B_p(\mathbf{x})$ to denote the continuous distribution over the location of landmark $p$ in *metric* (GC) space (its *belief*). Biophysically however, this would be encoded in the synaptic associations between PC $p$ and the GCs, i.e. the $p^{th}$ column of the matrix $\mathbf{B} \in \mathbb{R}^{N_P \times N_G}$. We consider these synaptic associations to constitute the *metric* embedding of sensory experience (Fig. 4A).

## 3.2 Online localization and learning

Given a suitable representation of uncertainty and a known map, localization is achieved by a process of recursive Bayesian estimation (RBE), where a model-based prediction based on perceived movement is corrected by incoming sensory information (Fig. 1A).

**Movement update** The location distribution (grid module activity) from the previous time-step $G(\mathbf{x}_{t-1})$ is updated according to perceived movement given a transition model $T(\mathbf{x}_t|\mathbf{x}_{t-1}, \hat{\mathbf{u}}_t)$:

$$G'(\mathbf{x}_t) = \int T(\mathbf{x}_t|\mathbf{x}_{t-1}, \hat{\mathbf{u}}_t) \cdot G(\mathbf{x}_{t-1})d\mathbf{x}_{t-1} \tag{1}$$

where $G'(\mathbf{x}_t)$ is the *movement estimate* and $\hat{\mathbf{u}}_t \sim \mathcal{N}(\mathbf{u}_t, \sigma_{\text{PI}}^2\mathbf{uI})$ is the noisy perceived movement at time $t$, where $\sigma_{\text{PI}}$ scales the noise with distance travelled. Since the firing of GCs are periodic across space, we use a wrapped Gaussian function defined over the triangular lattice to account for

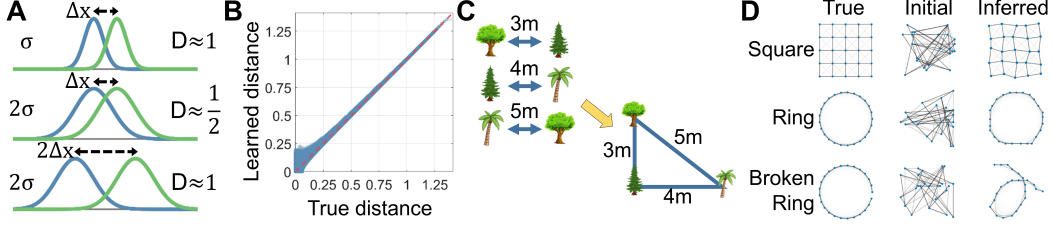

Figure 2: The *offline* model. **A** Inferred distance is a function of the 'overlap' in the receptive fields. **B** Inferred pairwise distances between place fields. **C** Inferred distances are used to recover the absolute structure of the world. **D** Structural inference on static structures with noisy initial priors (*"Initial"*). Inferred structure is sensitive to the topology of the environment (*"Broken Ring"*).

the probability of having transitioned from any of the infinite grid tilings:

$$T(\mathbf{x}_t|\mathbf{x}_{t-1}, \hat{\mathbf{u}}_t) = \sum_{m,n=-\infty}^{\infty} f(\mathbf{x}_t - \mathbf{x}_{t-1}|\hat{\mathbf{u}}_t + \mathbf{c}_{mn}, \sigma_{\mathrm{PI}}^2 \hat{\mathbf{u}}_t \mathbf{I}) \tag{2}$$

where and $\mathbf{c}_{mn} = 2\lambda(m\mathbf{v}_1 + n\mathbf{v}_2)$ is a spatial offset of scale $\lambda$ given the lattice basis vectors $\mathbf{v}_1 = [\cos(\phi), \sin(\phi)]$ and $\mathbf{v}_2 = [\cos(\phi + \pi/3), \sin(\phi + \pi/3)]$ and $\phi$ is the global orientation of the grid pattern. Where grid space is represented discretely by the firing rates of a population of GCs, the periodic form of the transition function can be replaced by multiplication by a velocity dependent circulant matrix $\mathbf{T}(\hat{\mathbf{u}}_t)$ (see Appendix C.1), linking to the eigendecomposition of diffusive transition matrices [51] and generalizing a previous mechanism to the case of noisy PI [5].

**Observation update** The *predicted estimate* is refined by incoming sensory input to give the *integrated estimate* $G(\mathbf{x}_t)$:

$$G(\mathbf{x}_t) = \frac{1}{K_t} H(\mathbf{p}_t|\mathbf{x}_t) \cdot G'(\mathbf{x}_t) \tag{3}$$

where $H(\mathbf{P}_t|\mathbf{x}_t) = \sum_{p=1:N_P} p_t^p B_p(\mathbf{x}_t)$ is the observation model defining the likelihood of the current sensory inputs $\mathbf{P}_t$ given the predicted location. The normalization constant $K_t$ is the sum over the current GC activity, implemented by a simple inhibitory feedback circuit: $\tau \frac{dG(\mathbf{x})}{dt} = -\int G(\mathbf{x})d\mathbf{x} + E$ , where $E = 1$ is a constant excitatory drive such that the sum of the GC activity sums to unity at steady-state.

**Online learning as prior formation** Where the distribution over landmark locations are encoded in the PC-GC synaptic weights matrix $\mathbf{B}$ and the predicted estimate of location by the GC firing rates $\mathbf{G}'_t$, a simple error-based learning rule with learning rate $\alpha = 1e-4$ minimizes the error between the *observation* and *movement* models (Fig. 1B):

$$\frac{1}{\alpha}\frac{d\mathbf{B}}{dt} = 2\mathbf{p}_t^\top(\mathbf{G}'_t - \mathbf{P}_t\mathbf{B}) \tag{4}$$

### 3.3 Offline message passing for probabilistic structural inference

During exploration of a novel environment, the *online* model produces stable learning when PI noise is low and the transition structure is static (Fig. 1A). However, all learning is local: only the synaptic weights of the currently active cells are modified at each time-step. This is not a full solution to the SLAM problem, which requires finding the most likely configuration of sensory observations (landmarks) $\{\mathbf{b}_p\}_{p=1:N_P}$ and current location (in grid space) given all historic observations and perceived movements, described by the joint *map-location* distribution $p(\mathbf{x}_t, \{\mathbf{b}_p\}_{p=1:N_P}|\mathbf{P}_{0:t}, \hat{\mathbf{u}}_{0:t}, \mathbf{x}_0)$ (see Appendix Fig. 1 for a summary of the anatomical mapping). Computing this requires integrating over all possible configurations of PC locations, which requires inference over previous and non-local observations. There are several advantages of a system capable of propagating information through non-local locations. Firstly, updates to the perceived location of a given landmark cause associated landmarks to also be updated without needing to be re-visited. Secondly, multiple weak (high variance) observations can together form strong hypotheses if those observations are consistent.

**The hippocampus as a cognitive graph**  The structure of an environment can be inferred from pairwise distance observations between landmarks [10; 40]. Intuitively, consider a 'spring network' of connected landmarks, where the edges represent noisy pairwise observations with stiffness and length equal to the certainty and estimated pairwise distance, respectively (see Appendix D.1). Convergence is contingent on the fact that, despite large absolute errors in landmark location (due to noisy PI), errors in relative pairwise distance measurements are correlated such that their variance decreases over time [10]. Relaxing the 'spring mesh' is equivalent to finding the maximally likely configuration of landmarks, if pairwise distance observations (pairwise potentials $\psi_{ij}$) are described by Gaussians with mean $d_{ij}$ and variance $\sigma_{ij} = \sigma^{PC} + d_{ij}\sigma^{PI}$ that are equal to and proportional to the perceived distance, respectively (the latter reflecting accumulation of PI noise in Eq. 2; Fig. 2C; Appendix D.1). The PC-GC synaptic associations can then be viewed as priors over the locations of each landmark in *metric* space, 'anchoring' the inferred structure which would otherwise be translation / rotation invariant. Together, the *associative* structure and *metric* mapping, encoded in the PC-PC (**A**) and PC-GC (**B**) associations respectively, define the posterior distribution over the landmark locations $\mathbf{b}_i$:

$$P(\{\mathbf{b}_p\}_{p=1:N_P}) = 2 \prod_{1 \leq i \leq N_P} \prod_{i \leq j \leq N_P} \psi_{ij}(\mathbf{b}_i, \mathbf{b}_j) \prod_{1 \leq i \leq N_P} B_i(\mathbf{b}_i) \tag{5}$$

where the $\psi_{ij}(\mathbf{b}_i, \mathbf{b}_j) = \psi_{ji}(\mathbf{b}_j, \mathbf{b}_i) = \sum_{m,n=-\infty}^{\infty} \exp\left(-\frac{1}{2}\sigma_{ij}^{-2}(d_{ij} - ||\boldsymbol{b}_i - \boldsymbol{b}_j + \mathbf{c}_{mn}||_2)^2\right)$ terms define the pairwise potentials between PCs and $\lambda$ is the grid scale. Note that $B_p(\mathbf{b}_p)$ here defines the continuous distribution of the location of PC $p$ in *metric* (GC) space for consistency with the literature; in reality it is a discrete vector described by the $p^{th}$ row of **B**.

*Associative* **encoding in the hippocampus**  We propose that these pairwise distance measurements are encoded in the recurrent synaptic associations between CA3 PCs, constituting an *associative* representation of the structure of space [40]. Given Gaussian place fields, a simple modified Hebbian learning rule with constant decay learns the pairwise PC weights (*associative* map) **A**:

$$\frac{1}{\alpha}\frac{dA_{ij}}{dt} = p_i(t)p_j(t) - A_{ij}^2 \tag{6}$$

which converges in the steady-state to $A_{ij} = \sqrt{< p_i(t), p_j(t) >}$, the square root of the correlation between the firing of two PCs (see Appendix D.1 for more details on the choice of learning rule). Where all place fields have equivalent receptive field covariance, the inferred Euclidean distance of PC $j$ from the perspective of $i$ is then proportional to the true distance given a simple transformation: $d_{ij}^2 = -log(A_{ij}) = (\mu_i - \mu_j)^2/2\sigma_{PC}^2$. The resulting form for the recovered distance is also scaled by the receptive fields' variance (the Bhattacharyya distance) [4], such that 'closeness' is related also to the 'discriminability' (Fig. 2A). We discuss this scaling constant later (see also Appendix D.3). Our approach differs subtly from typical graph-based SLAM systems [31; 57] which treat each observation independently. Instead, the CA3 synapses effectively average over multiple pairwise measurements. By assuming that noise in the pairwise distance measurements scale linearly with distance, both the mean and variance of the Gaussian describing this distribution is efficiently encoded in a single PC-PC synapse.

**Offline message passing for probabilistic structural inference**  The map configuration in Eq. 5 is approximated by message passing between PCs via the belief propagation algorithm (BP) [47], a single update cycle consisting of a message broadcast and a belief update. A message is defined as the probability distribution of a receiving node given the broadcasting node's belief and the pairwise potential $\psi_{ij}$ between the two (see below). Firstly, at iteration $n$ the node $t$ integrates all messages $m_{u \to t}^{(n)}(\mathbf{b}_t)$ received from its neighbours $u \in \Gamma_t$ with its prior self-belief $B_t^{(0)}(\mathbf{b}_t)$ to compute its updated self-belief:

$$B_t^{(n)}(\mathbf{b}_t) \propto B_t^{(0)}(\mathbf{b}_t) \prod_{u \in \Gamma_t} m_{u \to t}^{(n)}(\mathbf{b}_t) \tag{7}$$

Eq. 7 therefore represents the belief of node $t$ over its own state (location) given all messages from its connected neighbours in the graph and its prior. Secondly, node $t$ broadcasts messages back to its neighbours expressing its belief over their states:

$$m_{t \to u}^{(n+1)}(\mathbf{b}_u) \propto \int \psi_{tu}(\mathbf{b}_t, \mathbf{b}_u)B_t^{(n)}(\mathbf{b}_t)/m_{u \to t}^{(n)}(\mathbf{b}_t)d\mathbf{b}_t \tag{8}$$

where the new message is divided by the reciprocal message from the previous iteration [47].

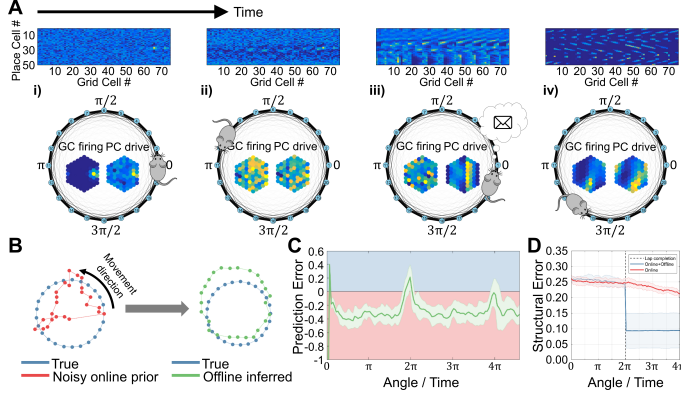

Figure 3: The *loop-closure* task. **A** The agent navigates a novel circular track, accumulating PI error. Lap completion (iii) triggers an *offline* inference event (see main text and Supp. Video 1) for details). **B** Structure inferred after loop-closure. **C** PE is reduced on completion of subsequent laps. **D** *Offline* inference allows one-shot learning when compared to the *online* system.

**Principled message scheduling** In a naiive 'sequential' schedule, all PCs broadcast messages before updating their beliefs. Instead, we implement an asynchronous message schedule ('Max-Entropy') in which only cells whose belief has changed by some threshold amount broadcast messages at the next time-step [11]. The 'message tension' $T_i^n$ is defined by the cumulative Jensen-Shannon divergence (symmetric KL; see Appendix E) between beliefs at successive time-steps: $T_i^n = T_i^{n-1} + \mathbb{JS}(\mathbf{b}_i^n || \mathbf{b}_i^{n-1})$. When the message tension is below a predefined threshold $T_{min}$, a node is considered converged and stops broadcasting messages. A single *offline* inference event is defined by the convergence of all nodes of the graph.

### 3.4 Prediction errors as an arbitration mechanism

Rather than continually perform map updates, we propose a more computationally (and energetically) favourable scheme in which the *offline* system is only recruited when the *online* system is performing poorly (batch updates are also known to be more robust [1]). We define the 'prediction error' (PE) of the *online* system: $\mathcal{E}_t = \mathcal{H}(\mathbf{G}_t') - \mathcal{H}(\mathbf{p}_t\mathbf{B})$, to compare the predicted and observed GC distribution, where $\mathcal{H}(\cdot)$ is the information entropy such that the PE term is positive when the inbound sensory information has a lower entropy than the current location estimate. *Offline* inference events are then initiated by positive PEs above a threshold $\mathcal{E}^0$. Note that the form of the PE update rule is similar but not identical to the rule for broadcasting messages during *offline* inference; sensory input that increases the entropy should not trigger *offline* inference events.

## 4 Results

### 4.1 Inference on static structures

We first tested the ability of the *offline* system to infer the structure of three environments (Fig. 2). Given erroneous initial estimates corresponding to priors formed during noisy PI, the system is able to correctly infer the true structures as those that satisfied pairwise measurements between states (Fig. 2D). However, an immediate consequence of the system is that this inferred structure will be sensitive to topology. Although PI will impose *metric* priors, where these priors are unreliable (as in the case of navigating around an unfamiliar ring environment under noisy PI), the inferred structure is sensitive to the 'closure' of loops (Fig. 2D, "*Broken Ring*").

### 4.2 Loop closure experiment

In the *loop closure* task (Fig. 3; Supp. Video 1), place fields are distributed uniformly around a circular 1D track. Initial location confidence is high, such that place and GCs active at the start location (0 *rads*) form strong associations. As the agent navigates around the track, PI error accumulates and the confidence in location decreases, resulting in subsequent PC-GC associations becoming more diffuse and less likely to correspond to the true structure (Fig. 3Ai). Due to the accumulated error, when the agent completes a full lap it receives a sharp input from the PCs initially active at the starting location, producing a strong positive PE and triggering an *offline* inference event (Fig. 3Aiii).

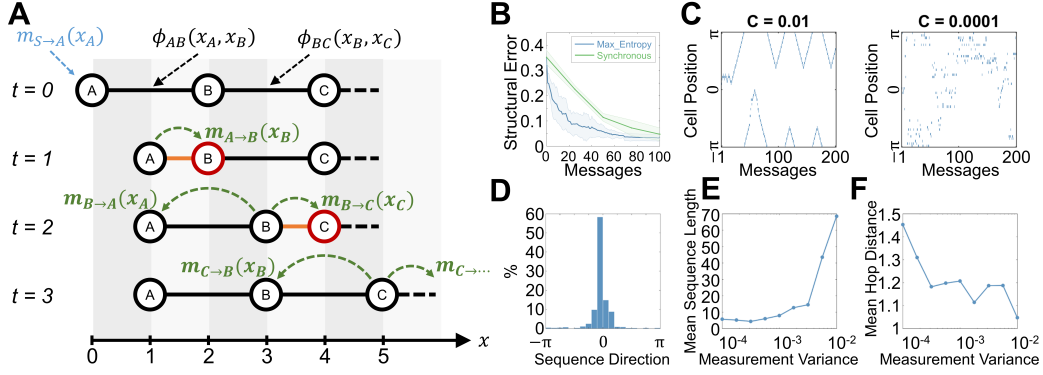

Figure 4: **A** Principled message scheduling generates PC sequences. Nodes are connected via their pairwise potentials $\psi$. (**t=0**) Sensory input causes an update to the belief of node $A$. (**t=1**) $A$ sends a message to $B$ causing it to update its belief. (**t=2**) Messages from $B$ only cause $C$ to update its belief, so only $C$ broadcasts at the next time-step. **B** The 'Max-Entropy' schedule converges faster than when all PCs broadcast messages at each time-step. **C** Examples of PC reactivation sequences. Multiple sequences occur simultaneously (Left) and become longer and smoother when pairwise measurements are less confident (Right; **E**, **F**). **D** Forward and reverse sequences occurred equally.

*Offline* **inference allows one-shot learning**    As expected, structural error is reduced significantly following the triggered *offline* inference events. This reduction is markedly larger than in equivalent trials using only the *online* system, resembling a 'one-shot' learning process (Fig 3D). Given the rapid map-learning, PEs on subsequent laps are also reduced (Fig. 3C).

**Principled message scheduling produces structured reactivations**    BP seeks a solution whereby messages received from neighbouring nodes cause negligible change to the receiving nodes' belief. The scheduling is therefore important from an energetic perspective; messages that do not produce changes in the beliefs of neighbours are redundant. Nodes which did not significantly update their self-beliefs following receipt of a message therefore do not need to re-broadcast a message at the next time-step (Fig. 4A).

In addition to the energetic advantages, the 'Max-Entropy' schedule also contributes to inference performance, converging faster than a simple 'sequential' scheme in which all nodes broadcast messages at each time-step, despite broadcasting fewer total messages (Fig. 4B).

The sequences of reactivations also contained significant structure, tending to propagate initially backwards along the track from the animal's current position, resembling the PC reactivations during reverse hippocampal replay (Fig. 4C) [15]. Sequences did not always hop to adjacent fields, occasionally hopping to new locations where remote sequences were then initiated (Fig. 4C) [8; 27; 53]. Multiple sequences at different remote locations can be seen to occur simultaneously or in an alternating fashion (Fig. 4C) [27]. Both forward and reverse sequences were observed in equal proportion (Fig. 4D) [15; 9]. Lastly, the 'hoppiness' of the sequences was related to the confidence in the pairwise observations, information propagating more quickly and smoothly in a 'stiffer' graph (a graph with more confident pairwise observations; Fig. 4E,F) [49; 27; 53].

### 4.3   Local distortions to the cognitive map

Grid patterns undergo significant local distortions in open environments, decreasing in scale and becoming less uniform (more sheared) towards the corners [23]. We hypothesized that these distortions might reflect the underlying structure of the environment as captured in the *associative* structure in CA3 and manifested in its projections to *metric* GC space.

In the same study, scale was also positively correlated to behavioural occupancy (animals spent more time in the middle of the environment; Fig. 5B, E bottom row, Appendix Fig. 2B) [23]. This effect was mirrored in our model, since over-sampling of the tails of the place fields near the boundaries of the environment led to the associated PCs overestimating their pairwise distances (Appendix Fig.

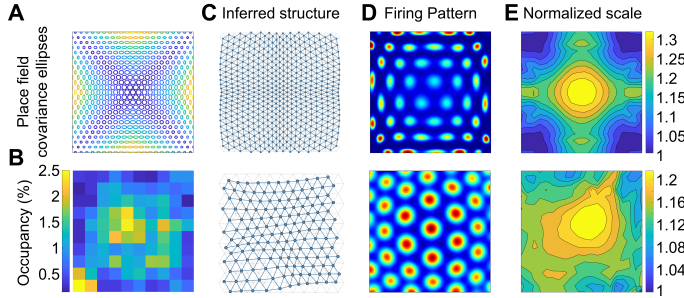

Figure 5: *Distortions to the cognitive map.* **A** Variation in place field shape results in distortions in the GC firing pattern (**D,E**, Top). **B** Learned distances due to biased sampling of the environment [23] also produce local distortions (**D,E**, Bottom). **C** Inferred structure in CA3.

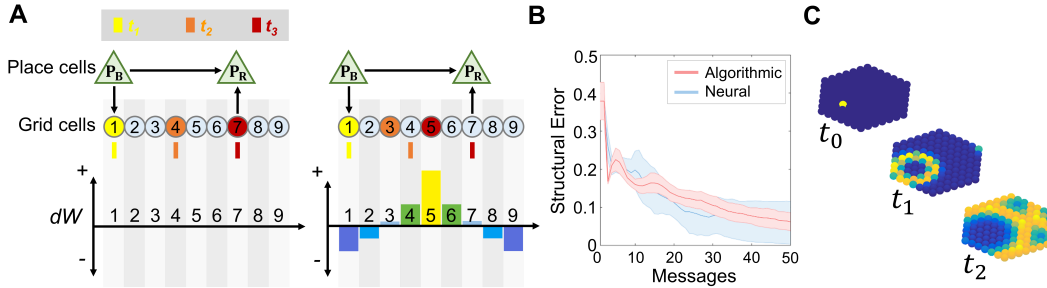

Figure 6: *A Neural model of coordinated HPC-mEC replay.* **A** (1D) The broadcasting PC $P_B$ sends a spike to neighbour $P_R$ (CA3), at the same time initiating a travelling wave in the GCs (mEC) by virtue of its synaptic projections **B**. (Left) No learning occurs when the spike and travelling wave arrive at $P_R$ at the same time. (Right) If the CA3 spike arrives ahead of the travelling wave, the synaptic associations of $P_R$ are adjusted towards the currently active GCs. **B** Comparison of the 'algorithmic' [32] and neural BP implementations. **C** Travelling waves on the 2D GC sheet.

2B); their mean co-firing was lower than expected if the animal were to sample from the place field uniformly; Fig. 2A). Note that pairwise *associative* distance is inversely related to the scale of the grid pattern readout, since larger *associative* distance implies travelling further in *metric* space (see Appendix D.1 and Appendix Fig. 2C)

Given that PC firing is related to the confidence of the presence of specific sensory cues, we also explored the case where place fields were sharper near the edges of an environment, as would be the case if driven by strong geometric cues (Fig. 5A, top row) [6; 25; 24]. These non-uniformities also produced the same local warping of the grid pattern Fig. 5C, D, E top row). This latter effect is attributed to the nature of Hebbian learning rules, whose learned synaptic strengths reflect the variance normalized distance between the fields, as opposed to the true Euclidean distance (Fig. 2A; Appendix D.1) [4]. Our model suggests that the cognitive 'distance' (or 'discriminability') between two sensory stimuli should be greater if the absolute confidence in the locations of each is greater.

### 4.4 A neural-level model of coordinated place-GC replay

How might belief propagation be implemented in the brain? More specifically, how might 'message broadcasts' correspond to spikes fired by PCs during replay, and how would GCs contribute to *offline* inference? Our proposed mechanism relies on coincidence detection by a 'receiving' PC $P_R$ of a direct spike from a 'broadcasting' PC $P_B$ and a travelling wave of activity across the GC population (Fig. 6A,C and Supp. Video 2).

A message broadcast is initiated by the firing of a spike from $P_B$ to synaptically connected PCs, with a transmission delay proportional to the inferred pairwise distance $d_{ij}$ (Fig. 6A, "Place cells"). In parallel, the same spike from $P_B$ drives activity in the GC population via the PC-GC synaptic associations (Fig. 6A, "Grid cells"). This activity propagates radially outwards at a constant speed, accumulating noise in proportion to the distance travelled (i.e. identically to PI during *online* localization; Fig. 6C; see Appendix D.2). This can be viewed as activity propagating through two generative models of *associative* and *metric* space (see Appendix D.3).

When $P_R$ receives the spike from $P_B$, we assume that the depolarization causes learning between $P_R$ and the currently active GCs, even though $P_R$ does not necessarily fire a spike [20; 18; 59]. If the distance indicated by the relative propagation of activity between GCs corresponding to the synaptic projections of $P_B$ and $P_R$ is equal to the distance encoded by the recurrent association between $P_B$ and $P_R$, $P_R$ will receive the spike from $P_B$ at the same time that the travelling wave arrives at the GCs to which $P_R$ projects, so that no significant synaptic changes are produced (Fig. 6A, Left). If these two distances are in disagreement, $P_R$ will revise its belief, shifting its synaptic associations to 'earlier' or 'later' GCs, respectively (Fig. 6C, Right).

Lastly, firing of $P_R$ is triggered only if there is significant change in its synaptic weights to the GC population, i.e. only messages indicating belief changes are propagated (a similar condition to that used to initiate the *offline* system). We propose therefore that the 'message tension' term, which governs spiking, might correspond to the accumulation of a learning related neuromodulator.

# 5   Discussion

During active exploration, place and grid cells are predominantly active when the location of the animal corresponds to their spatial receptive fields. During periods of rest or immobility however, the same cells have nonetheless been observed to reactivate at remote locations. That these distinct regimes of neural activity correspond also to distinct behavioural states suggests a functional role for *online* and *offline* processing in the HPC-mEC system.

Ours is the first model to demonstrate how PI and sensory based estimates could interact probabilistically during *online* localization in the HPC-mEC system (Fig. 4). We also show how more complex probabilistic inference could be performed via the *offline* interaction of HPC and mEC (Fig. 2, 6) and propose a detailed mapping of the joint *map-location* distribution to physiological correlates (Appendix Fig. 1). We then show how prediction errors between predicted and observed sensory stimuli can be used to efficiently arbitrate between the two systems (Fig. 3).

*Offline* inference events based on principled message passing resemble PC reactivations during replay events, which occur during offline behaviours such as pausing or sleep [15]. Our model predicts therefore that replay events (and associated sharp wave ripples) should be more frequent during structural changes to the environment rather than being solely responsive to reward [53], although rewards themselves may constitute salient sensory stimuli (an apple is highly indicative of location in an otherwise featureless maze). Our *algorithmic* and *neural* models [32] of this process are the first to predict the detailed interaction between PCs and GCs during coordinated replay events [45; 64]. Although investigated in a spatial context, structured information propagation may be a general mechanism for embedding *associative* experience in *metric* space [28; 19].

Our model is also the first to propose that observed local distortions to the grid pattern [23], reflect the underlying *associative* structure of the environment. Place fields are known to be smaller and more dense near to boundaries and salient locations [26]. Warping of the grid scale would thus be consistent with preserving a constant rate of change of sensory information [63]. The HPC-mEC interaction can be interpreted therefore as the embedding of *associative* structure within a *metric* map, to allow the agent to determine shortcuts between previously unexperienced state transitions [51].

**Acknowledgements**

We acknowledge funding from European Union's Horizon 2020 research and innovation programme Human Brain Project SGA2 (grant agreement no. 785907), Wellcome and ERC Advanced grant NEUROMEM.

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
