[Supplementary Material]

# 1 Acronyms

| | |
|---|---|
| $N_G$ | Number of grid cells |
| $N_P$ | Number of place cells |
| $\mathbf{x'} \in \mathbb{R}^2$ | Position in physical space |
| $\mathbf{x} \in \mathbb{R}^2$ | Position on the grid cell sheet (*metric* space) |
| $\mathbf{u} \in \mathbb{R}^2$ | Velocity in physical space |
| $\mathbf{G}_t \in \mathbb{R}_+^{1 \times N_G}$ | Vector of aposteriori grid cell firing rates at time $t$ |
| $\mathbf{G}'_t \in \mathbb{R}_+^{1 \times N_G}$ | Vector of predicted grid cell firing rates at time $t$ |
| $G(\mathbf{x})$ | Aposteriori distribution over location in *metric* space (continuous version of $\mathbf{G}_t$) |
| $G'(\mathbf{x})$ | Predicted distribution over location in *metric* space (continuous version of $\mathbf{G}'_t$) |
| $\mathbf{P} \in \mathbb{R}^{1 \times N_P}$ | Vector of place cell firing rates at time $t$ |
| $\mathbf{A} \in \mathbb{R}^{N_P \times N_P}$ | Matrix of PC-PC synaptic associations (the *associative* structure) |
| $\mathbf{B} \in \mathbb{R}^{N_P \times N_G}$ | Matrix of PC-GC synaptic associations (the *associative* embedding in the *metric* map) |
| $\mathbf{b}_i \in \mathbb{R}^2$ | Location of PC $i$ in *metric* space |
| $B_i(\mathbf{b}_i)$ | Belief of PC $i$ (continuous distribution of PC-GC synaptic associations) |
| $\psi_{ij}(\mathbf{b}_i, \mathbf{b}_j)$ | Pairwise potential (observation) between PCs $i$ and $j$ |
| $m_{u \to t}(\mathbf{b}_t)$ | Message from PC $u$ to $j$ |

# 2 Overview of complete system

## 2.1 Anatomy of SLAM

Our model proposes a detailed mapping of the joint *location-map* distribution to the HPC-mEC system:

Figure 1: Anatomy of a SLAM system. The joint *location-map* probability distribution (**A**) is represented in the firing rates and synaptic weights within the HPB-mEC system (**B**).

## 2.2 Algorithm describing the *online-offline* systems' interaction

| **Algorithm**: *Online* localization and learning with prediction error initiated *offline* inference | |
|---|---|
| 1:      initialize: $\mathbf{A} \sim \mathbb{U}(0,1), \mathbf{B} \sim \mathbb{U}(0,1)$ | % Initialize weight matrices |
| 2:    while $k < K$ do: | % For the duration of the simulation |
| 3:      do_movemement_update(); | % Update state via path integration |
| 4:      make_observation(); | % Compute PC firing |
| 5:      **if** $\epsilon_k > \epsilon_{min}$ **do**: | % If prediction error, do *offline* inference |
| 6:        **while** any $T_i < T_{min}$ **do**: | % Loop until cells below tension threshold |
| 7:        u=argmin(**T**); | % Find node with max. message tension |
| 8:        compute_and_broadcast_message(u); | % Broadcast message to neighbours of $u$ |
| 9:        **for** $t \in$ Neighbours$(u)$ **do**: | % Loop over neighbours of $u$ |
| 10:          update_belief(t); | % Update belief of node $t$ |
| 11:          update_message_tension(t); | % Compute change in belief od node $t$ |
| 12:        update_PC_GC_weights(); | % Do *associative* to *metric* map learning |
| 13:      update_PC_PC_weights(); | % Update *associative* map associations |
| 14:    do_measurement_update(); | % Correct state estimate |

# 3  *Online* model details

## 3.1  Neural implementation

In the movement *movement update*:

$$G'(\mathbf{x}_t) = \int T(\mathbf{x}_t|\mathbf{x}_{t-1}, \hat{\mathbf{u}}_t) \cdot G(\mathbf{x}_{t-1})d\mathbf{x}_{t-1} \tag{1}$$

the transition function is a wrapped Gaussian:

$$T(\mathbf{x}_t|\mathbf{x}_{t-1}, \hat{\mathbf{u}}_t) = \sum_{m,n=-\infty}^{\infty} f(\mathbf{x}_t - \mathbf{x}_{t-1}|\hat{\mathbf{u}}_t + \mathbf{c}_{m,n}, \sigma_{\mathrm{PI}}^2\hat{\mathbf{u}}_t\mathbf{I}) \tag{2}$$

$$= f_{\mathcal{W}}(\mathbf{x}_t - \mathbf{x}_{t-1}|\hat{\mathbf{u}}_t, \sigma_{\mathrm{PI}}^2\hat{\mathbf{u}}_t\mathbf{I}) \tag{3}$$

Since the transition function is shared across all states in grid space and the wrapped Normal distribution is periodic, eq. 3 can be represented as a circular convolution:

$$G'(\mathbf{x}_t) = f_{\mathcal{W}} * G_t \tag{4}$$

When grid space is represented discretely by the firing rates of a finite population of grid cells, the circular convolution in Eq. 4 can be equivalently represented as multiplication by a corresponding circulant matrix, such that Eq. 4 can be further manipulated to give:

$$\mathbf{G}'_t = \mathbf{G}_{t-1}\mathbf{T}(\hat{\mathbf{u}}_t) \tag{5}$$

where $\mathbf{T}(\hat{\mathbf{u}}_t)$ is a circulant matrix, each row containing an offset copy of the vector defined by the wrapped Gaussian in Eq. 3. This formulation links the functional form of the movement update to a plausible neural implementation, the convolution operation in Eq. 3 becoming a weighted sum of projections to a separate population of 'shifter cells' (**?**). In this work we simulate the transition function using Eq. 3, although our simulations (not included here) demonstrate that it is accurately approximated for arbitrary movement velocities via a weighted superposition of weight matrices with set offsets.

## 3.2  Generating the grid cell firing pattern

The firing pattern of a given GC (Main Text Fig. 1A) is generated by plotting the activity of the GC against the location of the simulated agent in physical space. If there is a preserved 'bump' of activity on the GC sheet (in *metric* space), as this bump moves according to path integration a given GC will periodically become active and inactive. Thus, a larger grid scale corresponds to a smaller angular velocity on the GC sheet; conversely, a larger angular velocity will cause the activity bump to complete more 'laps' of the sheet for a given movement in physical space (more vertices in the readout grid pattern).

# 4  *Offline* model details

## 4.1  Associative encoding in the hippocampus

Our PC-PC learning rule is based on the simple Hebbian mechanism of co-firing between cells, with weight decay to prevent unbounded growth. We chose to implement weight decay as the square of the weight magnitudes as opposed to the absolute magnitude for mathematical simplicity. The resulting form of the steady-state synaptic weights is in our case equivalent to the Bhattacharyya distance **?**, which is used as a measure of the 'discriminability' or variance-normalized distance.

Note that the distance of the encoding of two PCs in *metric* space implies that the local scale of the readout grid pattern at their corresponding locations in physical space (Fig. 2C). If all points in physical space are perceived as nearby, one must 'travel further' in physical space before travelling

Figure 2: **A** Spring network analogy of the *associative* structure of an environment. Edge 'stiffness' is inversely proportional to the variance in the Gaussian observation. **B** Offline distortions. Left: Pairwise distances near the edges of the environment are overestimated due to undersampling when the agent preferentially explores the middle of an environment. Colours denote the distance of the pair of PCs $i$ and $j$ from the walls of the $1 \times 1$m environment $d_{wall} = \sum_{p=i,j} \frac{1}{2}(min(x_p, 1 - x_p) + min(y_p, 1 - y_p))$. Right: Resulting local scale is proportional to the occupancy. **C** When the grid scale is smaller than the size of structure being encoded, we can think of 'wrapping' the structure onto the grid sheet. Here, colours denoted different tilings of the base *metric* tile (the Voronoi region of a given grid cell).

one 'period' in the periodic *metric* space (i.e. on the GC sheet), equivalent to the distance between two vertices of activity in the grid pattern.

## 4.2 Details of neural model implementation

**Travelling waves in neural media** In simulations, the travelling waves in mEC are simulated explicitly by calculating the true probability distribution at each time-step (i.e. corresponding each radial distance given the wave speed). However, it is known that various neural media can support traveling waves **??**. Note that in the literature, the exact phenomenon which we describe are referred to as 'travelling pulses', rather than 'travelling waves', the latter describing an increasing region of excitation (i.e. a propagating disc, rather than annulus).

Existing travelling pulse solutions preserve the exact shape of the initial stimulus pattern. However, here we present a novel model of a travelling pulse which broadens with travelled distance, mirroring the accumulation of PI error with distance travelled in all directions simultaneously (there is no bearing/angular information encoded in the pairwise PC-PC observations). Our model is based on a simple mechanical analogue, as used to model water-waves:

$$\frac{d^2\mathbf{v}}{dt^2} = c^2 \nabla'^2 \cdot H[\mathbf{v}]_+ \tag{6}$$

where $c$ is the speed of wave propagation, $H[\cdot]_+$ is the Heaviside function and the spatial Laplacian operator $\nabla = (\frac{d}{dx}, \frac{d}{dy})$ is replaced by a 2D Gaussian filter with variance equal to the PI noise. It was found empirically scaling the wave celerity $c = \alpha c'$ was required to match the desired propagation speed $c$, where $\alpha \approx 0.3$. The *wave* solution matched the probabilistic radial propagation of activity in the *algorithmic* solution **?** (Fig. 3B). However, we do not provide a mathematical proof of their equivalence.

We note that the wave could also in principle be generated by the same circuitry as used to propagate GC activity during *online* PI. However, whereas in the case of PI in a given direction the 'shifter' matrix corresponds to an offset Gaussian, in this case it would correspond to a ring with variance proportional to its radius. Recursive convolution with this filter (multiplication by the equivalent circulant matrix) would produce a 'travelling wave' rather than 'pulse' solution. To produce the travelling pulse, transient inhibition would be required such that only the 'front' of the wave is active. Ongoing work is investigating whether the *offline* and *online* dynamics could plausibly exist within the same system. In theory, since replay of spatial sequences occurs at rates $\sim 20\times$ faster **?** than physical experience, its plausible that if the timescale of the inhibition is fast, it would not affect PI during *online* PI.

Figure 3: **A** Illustration of belief propagation. PC *A* receives messages from PCs *B* and *C*. Messages take the form of rings, describing a preferred distance about the current beliefs of **B** and **C** with variance reflecting the confidence in the observation. The intersection of the messages uniquely determines the location of *A* over time. Note that *A* will also be broadcasting messages back to *B* and *C*. **B** Propagating messages as travelling waves in mEC. A physiologically realisti simulation of travelling waves with a modified Laplacian diffusion kernel closely approximates the probabilistic propagation of activity, reflecting the accumulation of PI noise in the broadening of the wave front.

**Belief update**   The belief update **?** of PC $t$ given a message from $u$ formally requires division of the previous message from $u$ to prevent 'double-counting' the information. This would require previous messages to be encoded in the synaptic weights. Instead, we implemented an approximation ignoring this previous division. We found empirically that this did not adversely affect the performance of the system (Fig. 6B); indeed it is known that message passing mechanisms are robust to the exact form of the belief and message computations **?**.

## 4.3  A generative modelling view of coordinated HPC-mEC replay

The *offline* process of inferring the *metric* embedding of the *associative* structure in the neural mechanism can be viewed as the simultaneous propagation of activity through two generative models:

$\frac{dS}{da} = W(S, a)$   World model which describes the evolution of the true stimulus state $S$.

$\frac{dS'}{da} = T^C(S, a)$   CA3 generative model which predicts the next stimulus state, where $S'$ is the predicted stimulus activity.

$\frac{dG'}{da} = T^G(G, a)$   mEC generative model which predicts the next grid state.

$G(S) = H(S)$   Mapping from stimulus to grid space (the observation model).

Note that 'generative' in this context corresponds to the transition function defined in the main text, and that we distinguish between the CA3 and mEC generative models with the superscripts $C$ and $M$, respectively. The aim of the system is to minimize two quantities. The first is to match the *associative* (CA3) generative model to the true observed stimulus transitions.

The second is to match the *metric* (mEC) generative model to the observations made by thee mapping from stimulus to *metric* space. We can think of this as minimizing the cost-function:

$$C = \left\lVert \frac{dS'}{da} - \frac{dS}{da} \right\rVert^2 + \left\lVert \frac{dG'}{da} - \frac{dG}{da} \right\rVert^2 \tag{7}$$

$$= \left\lVert \frac{dS'}{da} - \frac{dS}{da} \right\rVert^2 + \left\lVert \frac{dG'}{da} - \frac{dH}{dS}\frac{dS}{da} \right\rVert^2 \tag{8}$$

$$\tag{9}$$

The simplest analysis of this system corresponds to case where all terms are linear:

$$T^C(S, a) \quad = c \qquad\qquad c \text{ is a learned constant determining rate of change of location in the } \textit{associative} \text{ generative model for a given unit of action } a.$$

$$W(S, a) \quad = g \qquad\qquad g \text{ is the movement gain, a fixed property of the world dynamics that determines the true rate of change of the sensory stimulus.}$$

$$T^G(S, a) \quad = \alpha \qquad\qquad \alpha \text{ is the movement through the } \textit{metric} \text{ generative model for a given unit of action a.}$$

$$G(S) \qquad = K_0 S + K_1 \qquad \text{A linear projection from stimulus to } \textit{metric} \text{ space, where } K_0 \text{ and } K_1 \text{ correspond to the grid scale and spatial offset, respectively.}$$

The system above could represent an agent moving along a 1D track with a speed proportional to the 'action' $a$ ($g$ determines how far the agent travels through physical space for a given unit of action $a$) and current position $S$. The job of the generative model is then to match the intrinsic propagation speed $c$ (for a given action magnitude) to the rate of change of stimulus.

Secondly, we assume a linear mapping from stimulus $x'$ to *metric* space $x$. Note that, if the GC activity were to be driven purely by this linear observation model, increasing $K_0$ here would decrease the grid scale of the GC firing pattern and changing $K_1$ will determine its spatial offset. As in the *associative* generative model, the job of the *metric* generative model is to match simulated progression through *metric* space to the true progression as indicated by the observation model. Substituting the above into the cost-function, we get:

$$C = ||c - g||^2 + ||\alpha - K_0 g||^2 \tag{10}$$

Since $g$ is a fixed property of the world dynamics, the *associative* generative model will learn $c = g$, and the observation model $K_0 = \alpha/g$. Note that in the above case, since out models are linear we are making the explicit assumption that distances between stimuli are Euclidean in the *associative* generative model. $c$ therefore represents the scaling of this Euclidean metric and relates to the scaling constant outlined in the *associative* Hebbian learning rule in Section **??**. We assume that this constant is a physiological property of the network.

### 4.4 Analysis of replay sequences

States in the loop closure task were defined as angles around the track. For a single sequence $\bar{L} = \Theta_s = \{\theta_1^{(s)}, ..., \theta_{N_s}^{(s)}\}$, the mean sequence length was defined as $\frac{1}{S}\sum_{s=1:S} N_s$. The mean hop distance was defined as:

$$\bar{H} = \frac{1}{S}\sum_{s=1:S}\sum_{i=2:N_s}\frac{1}{N_s}abs(|\theta_i^{(s)} - \theta_{i-1}^{(s)}|_c) \tag{11}$$

where $|\cdot|_c$ is the circular distance (minimum of the clockwise and anticlockwise distances). Sequences were separated within a single *offline* inference event by defining a maximum hop distance $H_{max} = 5$.

## 5 Jensen-Shannon divergence

$$T_i^n \quad = \quad T_i^{n-1} + \mathbb{JS}(\mathbf{b}_i^n || \mathbf{b}_i^{n-1}) \tag{12}$$

$$= \quad T_i^{n-1} + \frac{1}{2}\Big[\mathbb{KL}(\mathbf{b}_i^n || m) + \mathbb{KL}(\mathbf{b}_i^{n-1} || m)\Big] \tag{13}$$

where $m = \frac{1}{2}(\mathbf{b}_i^n + \mathbf{b}_i^{n-1})$.