[Reviews · NeurIPS 2019]

Reviewer 1



-bravo for detailing computational mechanisms that may link associative processing in hippocampal place cells and metric processing in entorhinal grid cells -it is somewhat unclear what type of analyses and results have actually been aimed at and obtained - page 2: CA3 synapses are mentioned first summary of contributions -discussion: reward, in the eyes of this reviewer, should not be considered a 'sensory stimulus'; rather, reward processing is the result of neural processing of sensory cues in the environment -the figures and their captions are well thought out and effective in communicating their message

Reviewer 2



I'm mainly going to comment on the execution of the paper since I'm currently not very knowledgeable in the computational neuroscience of navigation in the brain: -Although it is easy to understand the paper content at a high level, I found it quite difficult to understand some important details, requiring multiple passes over the text to make sense of them. Examples: i) There are non-bold letters that denote continuous distributions over space (G, P), and boldfaced versions of them that represent "discretized" vectors that are grid and place cell responses. Is this mapping a simple discretization of the support of the probability functions? If not, what is the mapping? I guess this is a discretization at landmark locations for place cells (one landmark per place cell). Is it the same thing for the grid cells? Suggestion: Why don't you define the mapping in the beginning, and just use the discretized variables using simple matrix algebra? ii) It is a little hard to follow what is a scalar function and what is a vector. I'm assuming bold-faced letters are vectors and others are scalars? On the other hand, for instance T(x_t | x_before, u_t), which is non-bold, is defined as a sum of multivariate normals. iii) Use of \mathcal{N} is confusing. Do you define this to be a symbolic distribution to sample from, or a probability density function? Some mentions of \mathcal{N} have two arguments while some have three (possibly meant as a PDF). Please be consistent in your notation. I'd like to point out a particular use on line 75, because I was not able to understand the representation of a place cell: p_t^p=\mathcal{N}(x_t', \mu_p, variance). Do you define each place cell to represent the PDF of the normal distribution (if so, how)? Or do you mean each place cell is sampled from a normal distribution? Additionally, what does each of the three arguments represent? Is the mean x_t', or \mu_p? iv) I can understand that both the offline and the online update is for ultimately learning the mapping B from place cells to grid cells. However, the information flow is a little convoluted, and makes it a bit unclear what exactly the suggested mechanism is about. I believe that the methods section could be simplified with a more hierarchical organization, e.g. first emphasizing the formation of metric space through path integration (G') and associative map (PB), then combining the two for online localization, and then explaining the realtime and the offline update of B. - The contributions seem to be in the computational neuroscience of navigation; isn't there any related work or prior art that this work shares similarities with / builds on? (Given its popularity, we know that there is substantial work on the topic) For instance, what are some other proposed HPC/MEC related mechanisms that can predict the coordinated HPC-MEC replay? What some other models of combining path integration with place-related sensory signals for navigation? It is difficult to gauge the significance of this work without more context around the current state of knowledge in the field of computational neuroscience. I find this to be an important shortcoming. - The computational models of navigation that I'm familiar with (although admittedly I'm anything but up-to-date on the matter) typically depict HPC as mainly downstream from the cortex, creating the cognitive map through help from MEC. This is based on and corroborated by physiology data (see e.g. [1]). Things might have changed, but what is the biological motivation for proposing your circuit (with HPC->MEC connections most emphasized)? Given that multiple models can be proposed to account for physiologically observed data, it'd be more convincing to provide clear empirical grounds for the proposed model; doing so would strengthen the paper contribution. Minor comment: In eqns. (3) & (4), there is a small-case p_t , I guess this is a typo. [1] Zhang, Sheng-Jia, et al. "Functional connectivity of the entorhinal–hippocampal space circuit." Philosophical Transactions of the Royal Society B: Biological Sciences 369.1635 (2014): 20120516.

Reviewer 3



The paper tackles a very important problem and bridges two growing fields. The methodology is sound and different observed phenomena are covered to test the feasibility of the framework. The main problem of the paper is that its theoretical contribution is not explicitly specified. Particularly, some ideas in section 2 seems to be borrowed directly from the AI. While the authors have cited the related work in the beginning of the paper, I think it is necessary that they specify the source of each equation (or if it is from themselves).

[Author Response · NeurIPS 2019]

Dear all, thank you in earnest for the detail provided in each of the reviews. We have addressed in as much detail as
possible what seem to be the major points. Where we have not replied to a specific point, please assume that we agree
and will correct in the final manuscript.

*1. Motivation for work* We **agree that a more detailed discussion/broader citations of existing work** will strengthen
the case for our proposed model (R1-3). As noted (R2), there exists much excellent work modelling spatial processing.
Specifically, work implementing SLAM in a biological context (Milford et al., 2004; 2008), posing GCs as an
eigendecomposition of the transition structure of the environment (Stachenfeld et al., 2014; 2017; i.e. a HPC to GC
mapping (R2); **we will also include relevant experimental citations**) and proposing a method for correction of GC
activity by sensory inputs (e.g. Fuhs and Touretzky, 2006) we consider particularly relevant. However, **none of these**
**works discuss explicit probabilistic processing or representations**, which is clearly important to performing robust
inference under uncertainty. Ours is also the **first model of distortions** to the grid pattern, a subject of intense current
interest to the field (Krupic et al., 2018; Hagglund et al., 2019). If accurate, our model would be a significant advance in
the understanding of how the brain encodes space.

Neither do these models link to replay. Of existing models, **none account for coordinated mEC-HPC activity (R2)**,
only one gives normative insight (Mattar and Daw, 2018) and almost all focus on reward processing. We believe the latter
point emphasizes an important contribution of our work; recent work casts some doubt on the strength of the specific
predictions of RL based hypotheses (e.g. Stella et al., 2019). Not only does our theory pose an important alternative,
but it makes testable predictions. As correctly noted (R1), we are explicitly making the point that rewarded locations
are also likely to be locationally informative; an apple is highly indicative of location in an otherwise featureless maze.
We are not claiming that reward is an explicit sensory stimulus; **we agree that this point is clumsily made in the text**.
Lastly, that replay is observed in behavioural states such as sleep, immobility or at choice points in decision making
tasks, **implies that animals must rely on another mechanism for online localization (R1)** - as we propose.

*2. Contribution to AI* More importantly perhaps, we believe that our work is relevant to the AI community as a technical
"bridging" (R3) document and **aligns with several strands of active research**, in particular the graph network (GN)
community. It has been suggested that grid cells represent and eigendecomposition of the Laplacian of a graph-like
representation of environmental states (Stachenfeld et al., 2014). Take together with our work, this suggests that the
brain might perform hierarchical asynchronous message passing on low-dimensional embeddings of such a graph.
Whereas there have been several recent paper on learnable attention modules (Velickovic et al., 2018) and other methods
for reducing the complexity of graph convolutions (e.g. Bruna et al., 2014; Kipf and Welling, 2017), to our knowledge,
there is no existing work on asynchronous processing in GNs (despite the success of the mechanism employed in our
paper, although originally proposed by Elidan et al., 2012, which **we will make more clear where relevant (R3)**). We
would hope to stimulate serious discussion, given evidence for these processes in the brain and **supporting theory**
(our work and others), as to whether these approaches are worth investing more energy in investigating, especially
given current trends in neuromorphic hardware and the increasing need for low-power solutions in e.g. mobile devices.
Notwithstanding the above, and although we do not believe these are the primary contributions of this paper, to
our knowledge it is the first to detail mathematically the process of performing belief propagation (BP) in **periodic**
**manifolds with hexagonal** symmetry (R3; there are clear cases where this may be better than using square bases, c.f.
Hoogenboom et al. 2018), or to use principled message scheduling to selectively update the map in a SLAM context.

*3. Outline of contributions* Given the above, we will improve the introduction to **a. State more clearly what results**
**are obtained and what analyses are performed (R1/3)** and **b. Outline their relevance the AI/Neuro fields (R3)**.

*4. Mathematical notation* We thank R2 for their detailed comments. **We will correct what we agree is abusive use** of
*mathcal{N}* and apologise if it caused difficulty in interpreting the methods. Likewise regarding the mixed usage of
scalar functions and vectors. Initially, we felt it important to abstract as much as possible the technical details from their
mechanistic implementation. On re-reading however, we appreciate that it causes the notation to become unnecessarily
unwieldy. **As suggested, we will specify the mapping and continue therein with linear algebra notation**.

*5. Structure of methods* We thank R2 for their suggestion and **will restructure if it will improve the clarity of the**
**work**. To be clear, we do not consider how the metric space (i.e. the GC-GC connectivity) is formed (although c.f.
Stachenfeld et al., 2014; Whittington et al. 2018). Rather, the metric space is fixed, but the associative map (PC-PC
connections) and the associative-metric mapping (GC-PC) are learned. Our hypothesis is that, given no sensory drive
to the grid cells, the agent has some weak prior notion of 'path integration' that would be based on self-motion cues,
which is used by the *online* system to form priors over the encoding of encountered landmarks. The role of the *offline*
update is then to correct this encoding, given the joint distribution over all landmarks. In this way, separated clusters of
associated landmarks are still embedded proximally in the metric space by virtue of their encoding via path integration.
The distortions arise from the notion that the self-motion estimate of 'distance travelled' through metric space is only
approximate; in fact, we believe that it is affected by information flow, which is captured in the pairwise landmark
distances (PC-PC connections) and ultimately induces distortions in the readout.

[Meta-Review · NeurIPS 2019]

The authors present a model of localization and mapping in the place/grid-cell circuits of hippocampus and entorhinal cortex. The paper is polished and well-written, though it requires a fair background in neuroscience, making it difficult for the average NeurIPS reader to appreciate. More importantly, the relationship to previous work is poorly explained. Still, I suggest that this paper be accepted, with the condition that the authors follow through on their promise to improve the exposition and include an explicit related works section.